# Interaction and Collaboration of SP1, HIF-1, and MYC in Regulating the Expression of Cancer-Related Genes to Further Enhance Anticancer Drug Development

Kotohiko Kimura, Tiffany L. B. Jackson ⓘ and Ru Chih C. Huang *ⓘ

Department of Biology, Johns Hopkins University, 3400 N. Charles Street, Baltimore, MD 21218-2685, USA
* Correspondence: rhuang@jhu.edu; Tel.: +1-410-516-5181; Fax: +1-410-516-5213

**Abstract:** Specificity protein 1 (SP1), hypoxia-inducible factor 1 (HIF-1), and MYC are important transcription factors (TFs). SP1, a constitutively expressed housekeeping gene, regulates diverse yet distinct biological activities; MYC is a master regulator of all key cellular activities including cell metabolism and proliferation; and HIF-1, whose protein level is rapidly increased when the local tissue oxygen concentration decreases, functions as a mediator of hypoxic signals. Systems analyses of the regulatory networks in cancer have shown that SP1, HIF-1, and MYC belong to a group of TFs that function as master regulators of cancer. Therefore, the contributions of these TFs are crucial to the development of cancer. SP1, HIF-1, and MYC are often overexpressed in tumors, which indicates the importance of their roles in the development of cancer. Thus, proper manipulation of SP1, HIF-1, and MYC by appropriate agents could have a strong negative impact on cancer development. Under these circumstances, these TFs have naturally become major targets for anticancer drug development. Accordingly, there are currently many SP1 or HIF-1 inhibitors available; however, designing efficient MYC inhibitors has been extremely difficult. Studies have shown that SP1, HIF-1, and MYC modulate the expression of each other and collaborate to regulate the expression of numerous genes. In this review, we provide an overview of the interactions and collaborations of SP1, HIF1A, and MYC in the regulation of various cancer-related genes, and their potential implications in the development of anticancer therapy.

**Keywords:** hypoxia-inducible factor 1; SP1; MYC; cancer

## 1. Introduction: Specificity Protein 1, Hypoxia-Inducible Factor-1, and MYC as Master Regulators of Cancer

Recent progress in systems biology has shown that several specific factors are participants of a network that function as master regulators of cancer [1–3]. Wilson and Volker Filipp investigated complementary omics in human cancer, and discovered a close teamwork of transcriptional and epigenomic machinery, which is tightly connected and comprises histone lysine demethylase 3A, basic helix-loop-helix factors, MYC, hypoxia-inducible factor 1 alpha (HIF1A), and sterol regulatory element-binding transcription factor 1, as well as differentiation factors such as activator protein 1, myogenic differentiation 1, specificity protein 1 (SP1), Meis homeobox 1, zinc finger E-box-binding homeobox 1, and ETS like-1 protein (ETS1) [1]. Cao et al. [2] showed that 10 long non-coding RNA (lncRNA)-transcription factor (TF) pairs including four glycolysis-related lncRNAs (FTX, long intergenic non-protein coding RNA 472, proteasome 20S subunit alpha 3 antisense RNA 1, and small nucleolar RNA host gene 14) and six TFs (forkhead box protein P1, SP1, MYC, FOX-M1, hypoxia-inducible factor 1 alpha [HIF1A], and FOS) are involved in the progression of human lung adenocarcinoma. Malik et al. [3] discovered, using a statistical method called CoMEx (Combined score of DNA Methylation and Expression) to assess differentially expressed and methylated genes/microRNAs (miRNAs) between human seminoma and normal tissues, two hub miRNAs (miR-182-5p and miR-338-3p), five hub

TFs (ETS1, HIF1A, hepatocyte nuclear factor-1 alpha, MYC, and SP1), and three hub genes (*cadherin 1*, *C-X-C chemokine receptor type 4*, and *Snail family transcriptional repressor 1*) in the seminoma-specific regulatory network. Interestingly, in all of these studies, three TFs, namely SP1, HIF1A, and MYC, were among the factors that participated in the cancer regulatory network. In addition, many studies have shown that SP1, HIF1A, and MYC are often upregulated in cancer [4–9]. Together, these data suggest that SP1, HIF1A, and MYC have crucial roles in cancer development, and that interfering with their activity could negatively impact cancer development and progressions. For this reason, enormous efforts have been undertaken to develop inhibitors for SP1, HIF1A, and MYC. Accordingly, numerous inhibitors of SP1 or HIF1A have been developed [10–14]; however, designing MYC inhibitors has been extremely difficult [15]. Nevertheless, all of the inhibitors against SP1, HIF1A, or MYC can be considered potential anticancer drugs due to the nature of these TFs as master regulators of cancer.

## 2. What Are SP1, HIF-1, and MYC, and How Do These TFs Benefit Cancer

SP1, HIF-1, and MYC are three major TFs that play important roles as master regulators of cancer, so the next question is—what are these TFs and how do they benefit cancer as regulators of gene expression?

### 2.1. SP1: Housekeeping Gene That Regulates Biological Activities

SP1 is a ubiquitous TF from the Sp/Krüppel-like family (KLF) of TFs, which are the major forms of zinc finger DNA-binding proteins [16]. The defining feature of SP1-like/KLF proteins is a highly conserved DNA-binding domain (>65% sequence identity among family members) at the C-terminus that has three tandem Cys2His2 (C2H2) zinc finger motifs [17]. Likewise, SP1 contains three highly homologous C2H2 regions [18,19], which exhibit direct binding to DNA at the C-terminal regions of the protein, thus enhancing gene transcription [20]. By contrast, the N-terminal regions of the proteins are more divergent [21]. SP1 has four unstructured domains A, B, C, and D, starting from the C-terminal region of the protein. The two main transactivating domains of SP1 are A and B, which are capable of direct interaction with the components of transcription machinery such as TATA-binding protein (TBP) and TBP-associated factor 4 [22]. The C domain is not indispensable but is highly charged and supports DNA binding and transactivation. The D domain, also known as the C-terminal region of SP1, has multimeric domains and is responsible for the binding of consensus sequences such as 5′-(G/T) GGGCGG(G/A)(G/A)(G/T)-3′ (the sequences are referred to as the GC box) [23]. The N-terminal region of SP1 is a small inhibitory domain, which mainly regulates the functions of domains A and B and is linked to the A domain with a serine/threonine-rich region [22]. The transcriptional activity and stability of SP1 are influenced by its post-translational modifications. SP1 undergoes acetylation, SUMOylation, ubiquitination, and glycosylation after its translation [24,25]. Acetylation of SP1 takes place in the DNA-binding domain [26]. Glycosylation occurs at the O-GlcNAc linkages at the serine and threonine residues in SP1, which can either induce or suppress DNA binding and transcription [27]. SUMOylation, occurring in the Lys16 region, controls the transcription of SP1 by instigating alterations in the chromatin structure, making the DNA inaccessible for transcription [28]. The proteasomal degradation of SP1 is carried out by ubiquitination, where the β-transducin repeat-containing protein (TCRP) ubiquitin ligase complex interacts with SP1 through the DSG (Asp-Ser-Gly) destruction box (β-TCRP binding motif) within the C-terminus of SP1 [29]. SP1 is critical for early embryonic development [30,31], but its expression decreases with age and there is evidence that the transformation of normal cells to cancer cells is associated with the upregulation of SP1, SP3, and SP4 [10,32]. Functional studies have demonstrated that the SP-like family of TFs regulates various genes responsible for cancer-related cellular mechanisms; SP1, SP3, and SP4 are also non-oncogene addiction (NOA) genes and thus are important drug targets [33]. NOA genes are essential for supporting the stress-burdened phenotype of tumors and thus are vital for their survival. The most important functional role of SP1 in normal cells is

the regulation of cell cycle and cellular reprogramming [5]. Since cell proliferation and differentiation are the most active during the developmental stage of organisms, SP1 plays critical roles during early developmental stages perhaps for this reason [30,31]. This also indicates that SP1 is still an essential component of cellular mechanisms during adulthood although less so compared with during developmental stages.

### 2.2. HIF-1: Functions as a Mediator of Hypoxic Signals

HIF-1 is the most important factor involved in the cellular response to hypoxia [34,35]. The broad impact of HIF-1 on cell biology is reflected in the total number of hypoxic target genes, which is estimated to be approximately 1–2% of all human genes [36]. HIF-1 plays important roles in energy metabolism and angiogenesis, especially in cancer progression [34]. It is composed of two subunits, HIF1A and HIF1B (aryl hydrocarbon receptor nuclear translocator). Among these two subunits, only HIF1A is activated under hypoxia and HIF1B is not regulated by oxygen [35]. The dual functional protein apurinic/apyrimidinic endonuclease 1 is an enzyme in DNA base excision repair but also works as a redox factor to maintain HIF1A in the reduced state that is necessary for its transcriptional function [35]. In the presence of oxygen, prolyl hydroxylase hydroxylates HIF1A and hydroxylated HIF1A binds to the tumor suppressor von Hippel–Lindau protein (pVHL), a component of the E3 ubiquitin ligase complex. This interaction causes HIF1A to become ubiquitinated and targeted to the proteasome, where it is degraded. However, under hypoxia, HIF1A is not degraded by the proteasome since prolyl hydroxylase is not functional, so HIF1A dimerizes with HIF1B and binds to the hypoxia response element (HRE) in the promoters of target genes, initiating the expression of genes that promote adaptation to hypoxia [35]. HIF1A as well as the more cell-specific HIF2A are important regulators of the hypoxic response. Although both HIF1A and HIF2A are highly conserved at the protein level, share a similar domain structure, heterodimerize with HIF1B (HIF-2 is formed by the assembly of HIF2A and HIF1B), and bind to the same DNA sequence (the HRE), their effects on the expression of various genes differ [37].

### 2.3. SP1 and HIF-1

The importance of HIF-1 and SP1 in cancer development is beyond dispute. In fact, it has been shown that both HIF-1 and SP1 are involved in every aspect of cancer-related cellular mechanisms. For instance, both SP1 and HIF-1 play important roles in the regulation of cancer metabolism in carbohydrates [34,38–41] and lipids [42–44]. Both are involved in anticancer immunity via regulation of immune-related cells [45–51]; the tumor microenvironment (TME)/oncometabolites [52–58]; and transforming growth factor beta, which regulates the immune system [59–64]. SP1 promotes tumor angiogenesis via activation of vascular endothelial growth factor (VEGF), epidermal growth factor receptor (EGFR), and VEGF receptor 3 (VEGFR3) [65–67], whereas HIF-1 is a master regulator of angiogenesis, participating in vasculature formation by synergistic correlations with other proangiogenic factors such as VEGF, placental growth factor, and angiopoietins [68]. In addition, SP1 plays an important role in each of the crucial events of metastasis, namely, adhesion, invasion, migration, and angiogenesis [65–67,69–71]. Both SP1 and HIF-1 are also involved in the regulation of cellular stress mechanisms as mediators of the protection of cancer cells against various stresses [72–74].

### 2.4. MYC: A Master Regulator of Cellular Activity

MYC is a transcription factor that belongs to the basic helix-loop-helix-leucine zipper (bHLHZip) family and regulates cell growth, differentiation, metabolism, and cell death. Thus, MYC functions as a master regulator of major cellular functions [75–78]. Studies using knockout mice have shown that MYC is particularly important for cell growth (accumulation of the body mass) and is indispensable during the period of both embryogenesis and adulthood [79]. c-MYC is the prototype member of the MYC family, which also includes N-MYC and L-MYC proteins in mammalian cells. All three members of the

MYC family are highly homologous but distributed differently. c-MYC is ubiquitous and highly abundant in proliferating cells, whereas N-MYC and L-MYC display more restricted expression at distinct stages of cell and tissue development. MYC proteins exist within the MYC/MAX/MXD network. To fold and become transcriptionally active, c-MYC must first heterodimerize with MAX, a process governed by the coiling of their bHLHZip domains. Once dimerized, the c-MYC/MAX complex acts as a master transcriptional regulator by binding via its basic region to the specific DNA consensus sequence 5′-CANNTG-3′. Due to the multifunctional activities of MYC in cellular functions, cancers with MYC activation elicit many of the important hallmarks essential for autonomous neoplastic growth. In fact, MYC aberrations or upregulation of MYC-related pathways occur in many cancers. In preclinical animal models, MYC inactivation can result in sustained tumor regression, a phenomenon that has been attributed to oncogene addiction [80].

Recently, it was shown that MYC overexpression leads to increased chromatin interactions at super-enhancers and MYC-binding sites [81]. This shows the importance of MYC overexpression in the regulation of cancer development and suggests that super-enhancers might be a potential target for anticancer therapy. Recent studies have also demonstrated that MYC signaling can enable tumor cells to dysregulate the TME and evade the host immune response [82]. Due to the importance of MYC as a regulator of both cancer and the TME, MYC inhibitors may be a holy grail of anticancer drugs. For this reason, many therapeutic agents that directly target MYC are under development; however, to date, their clinical efficacy remains to be demonstrated partially due to the extreme difficulty of developing efficient MYC inhibitors specifically targeted for cancer therapy [15,83]. In this regard, Omomyc, a newly developed MYC inhibitor is more specific in targeting MYC-related genes responsible for cancer development than other MYC inhibitors, might provide insights into how to target MYC for cancer therapy [84].

## 3. Interactions among SP1, HIF-1, and MYC with One Another and Other TFs
### 3.1. Modulation of SP1, HIF-1, and MYC Activities

SP1, HIF-1, and MYC modulate the expression of numerous genes as major TFs. However, these TFs do not work independently and are in fact under the regulation of many other cellular components. For example, SP1, HIF-1, and MYC can interact and modulate the activities of each other. Figure 1 shows the promoters of human *SP1*, *HIF1A*, and *MYC* genes [85–92]. 'SP1' and 'HRE' in the figure indicate the locations of SP1 and the HRE consensus sequences, respectively. The SP1 consensus sequences are usually the GC boxes, whereas the HIF-1 consensus sequences (of the HRE) usually contain the nucleotide residues '5′-RCGTG-3′. The *SP1* promoter contains numerous SP1 consensus sequences as well as NF-Y and E2F consensus sequences. SP1 binds to NF-Y and E2F consensus sequences as well as SP1 consensus sequences in the *SP1* promoter [86,87]. These data suggest that SP1 can autoregulate its transcriptional activity. In addition to these consensus sequences, there is an HRE in the *SP1* promoter (Figure 1) [85] to which HIF-1 binds and stimulates the transcriptional activity of the *SP1* promoter [85]. It has been shown that the mRNA and protein levels of SP1 are decreased by silencing HIF1A in human cultured esophageal squamous cell carcinoma cells, whereas overexpression of HIF1A significantly increases these levels [85]. These data indicate that HIF-1 upregulates SP1 through its binding to the HRE.

There are numerous SP1 consensus sequences in the *HIF1A* gene promoter, which suggests that SP1 can induce *HIF1A* gene expression (Figure 1) [88,89]; however, no definitely active HRE has been found in the *HIF1A* promoter to date (Figure 1) [88,89]. These data confirm that the HRE in the *SP1* gene promoter contributes to the induction of *SP1* gene transcription by HIF1A, and SP1 consensus sequences in the *HIF1A* gene promoter contribute to the induction of *HIF1A* gene transcription by SP1 [85–89]. Thus, there may be a positive activation feedback loop of HIF-1 and SP1, and hypoxia-mediated induction of HIF-1 may trigger the activation of both SP1 and HIF-1 until normoxia deactivates HIF-1. On the other hand, in the promoter of the *MYC* gene, there is one SP1 consensus sequence

located upstream of the transcription start site (TSS) and two located downstream of the TSS [90–92]. Meanwhile, there are two E2F consensus sequences in the *SP1* gene promoter, to which MYC and SP1 can bind (Figure 1). Although, as discussed later, many studies have shown that SP1 and MYC collaborate in the transcriptional regulation of various genes, to date, there has been no definitive study showing that SP1 directly regulates transcription of the *MYC* gene [90–92] or that MYC directly regulates transcription of the *SP1* gene.

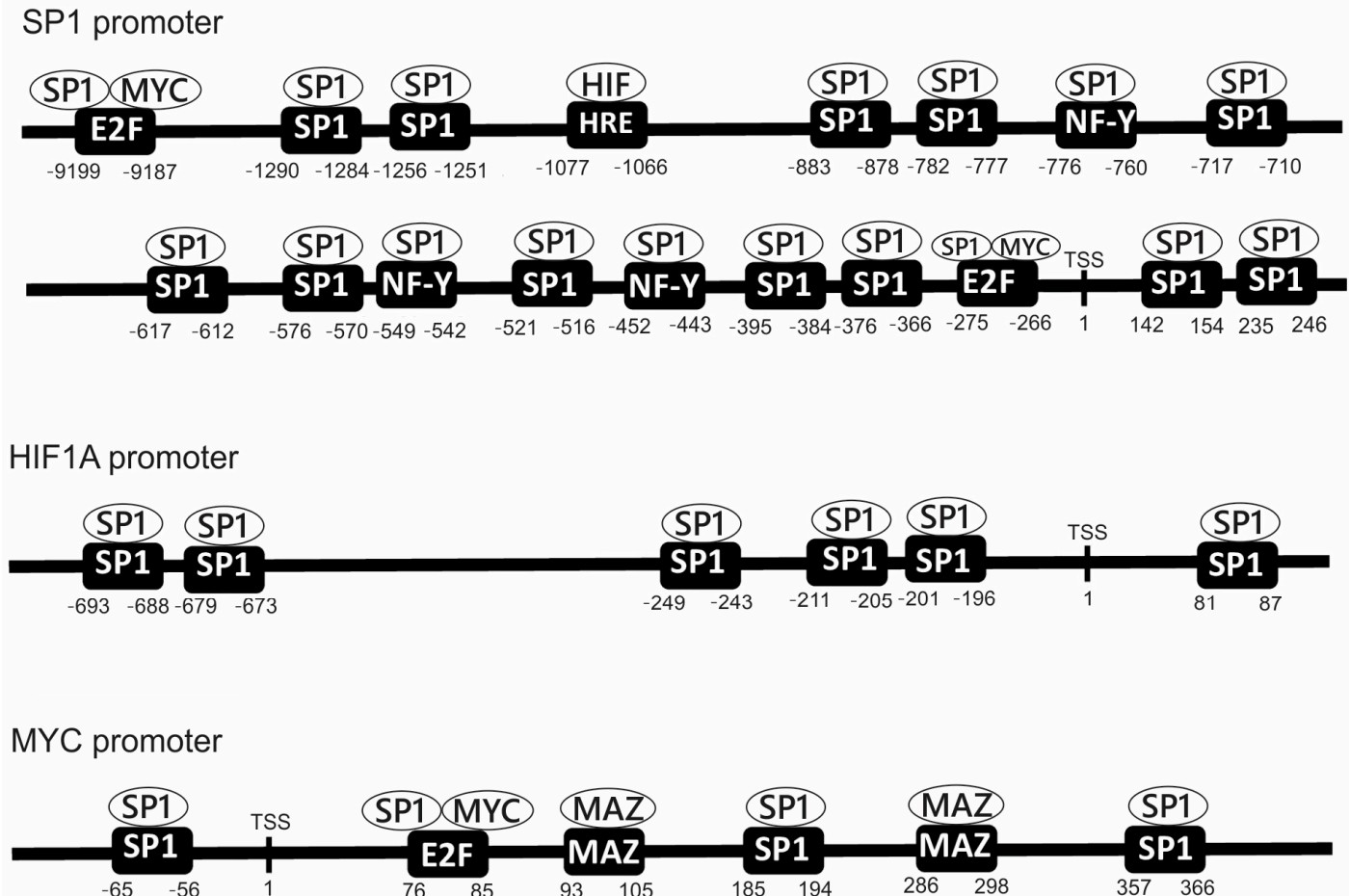

**Figure 1.** The promoter structures of human *SP1*, *HIF1A*, and *MYC* genes. The consensus sequences and their potential binding proteins are shown in each promoter. SP1 binds to SP1 consensus sequences (GC box) as well as NF-Y and E2F consensus sequences. HIF-1 binds to the HRE. MYC, similarly to SP1, binds to E2F. Myc-associated zinc finger protein (MAZ) is an important regulatory protein associated with *MYC* gene expression and binds to MAZ consensus sequences [90,92]. The nucleotide numbers are numbered from the transcription start site (TSS). The TSS for *MYC* gene promoter is for the P1 promoter [90,92].

### 3.2. Effect of HIF-1 on SP1 Gene Expression and Vice Versa

Expression of the *SP1* gene can be upregulated by HIF-1 transcriptionally by the binding of HIF-1 to its consensus sequences in the *SP1* gene promoter, as described in Section 3.1 [85]. This is shown schematically in Figure 2A. Meanwhile, Figure 2B–D schematically shows how *HIF1A* expression is regulated by SP1, using several examples. Insulin increases *HIF1A* promoter activity by reactive oxygen species (ROS) via SP1 in murine 3T3-L1 preadipocytes [93]. *HIF1A* transcription is downregulated by protein arginine methyltransferase 1 (PRMT1), a protein whose transcription is regulated by SP1 in human Hela cervical carcinoma and human HEK293T embryonic kidney cells [94]. In the former example, SP1 is activated by phosphoinositide 3-kinase/protein kinase C via ROS, and then induces *HIF1A* transcription (Figure 2B). In the latter example, the

suppression of PRMT1, which prevents the recruitment of SP1/SP3 to the *HIF1A* gene promoter, allows SP1/SP3 to activate the transcription of HIF1A (Figure 2C). In both cases, SP1 directly induces transcriptional activity of the *HIF1A* gene via its binding to SP1 consensus sequences in the *HIF1A* gene promoter (Figure 1) and upregulates expression of the *HIF1A* gene [77–89]. Meanwhile, SP1 can indirectly regulate HIF1A expression by modulating the gene expression of histone deacetylase 4 (*HDAC4*) in rat cardiomyocytes (Figure 2D) [95]. SP1 upregulates the activity of the *HDAC4* gene promoter, thereby promoting deacetylation and impairing the secretion of high mobility group box 1 in mouse intestinal epithelial cells [96]. Likewise, HDAC4 can prevent the acetylation of HIF1A, thereby stabilizing the protein in human pVHL-null kidney cancer cell lines [97]. In this way, SP1 upregulates HIF1A expression either directly by activating *HIF1A* gene expression via binding to the *HIF1A* gene promoter (Figure 2B,C) or indirectly by stabilizing HIF1A protein via modulation of *HDAC4* gene expression (Figure 2D). Either way, SP1 increases the activity of HIF1A. Unlike the *HIF1A* gene, the *HIF1B* gene is constitutively expressed [98]. Therefore, the activity of HIF-1, which is composed of HIF1A and HIF1B, is regulated by adjusting the mRNA and protein levels of HIF1A in cells as well as by modulating the levels of co-activators for HIF-1 [37].

## The mechanism of SP1 activation by HIF1A

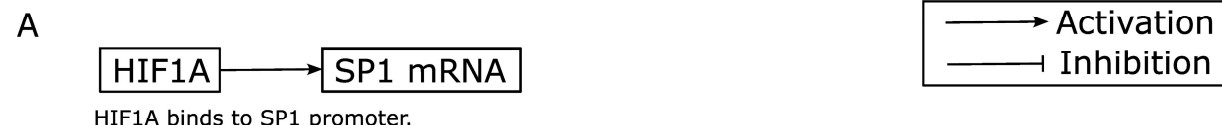

## The mechanism of HIF1A activation by SP1

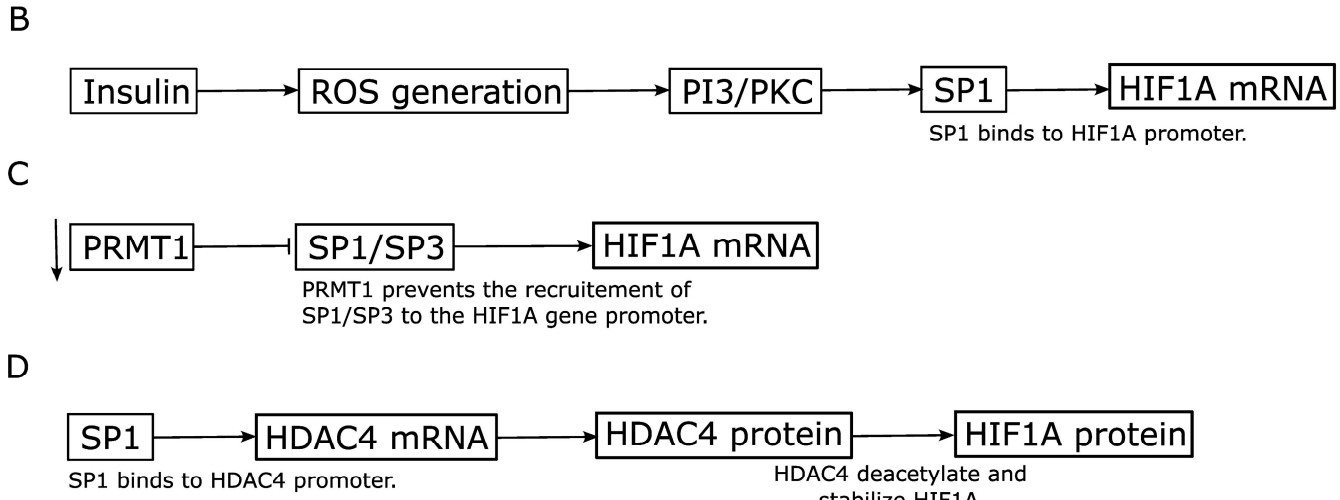

**Figure 2.** Schematic of the mechanism of activation of SP1 by HIF1A and that of HIF1A by SP1. (**A**) The mechanism of SP1 activation by HIF1A. (**B–D**) The mechanism of HIF1A activation by SP1. While HIF1A regulates SP1 expression via binding to the promoter of a *SP1* gene, SP1 can regulate HIF1A expression at both the mRNA level (**B,C**) and protein level (**D**). PKC: protein kinase C.

### 3.3. Effects of HIF-1 Compared to the Effects of SP1 on MYC Gene Activities

While HIF-1 induces *SP1* gene expression, it inhibits the activity of MYC (without affecting *MYC* gene expression) [85,99–101]. Since activation of MYC is usually associated with cell growth, MYC activities must be suppressed under hypoxia, which is a condition unsuitable for rapid cell growth due to a lack of oxygen, which is required for efficient biological energy production. Thus, under hypoxia, MYC activity is inhibited by HIF1A as

an adaptive response that promotes cell survival under low oxygen conditions. Since there is no HRE in the *MYC* gene promoter (Figure 1), HIF1A is unlikely to inhibit transcription of the *MYC* gene by directly binding to the *MYC* promoter. However, there are several mechanisms by which HIF can inhibit MYC activity. First, HIF1A can antagonize MYC transcriptional activity at MYC target genes by interfering with MYC binding to protein partners. For instance, HIF1A binds to MAX and disrupts MYC/MAX complexes, leading to reduced cyclin D2 expression, induction of p21 (CDKN1A), and G1 phase cell cycle arrest in human pVHL-null kidney cancer cell lines [102]. Meanwhile, under hypoxia, HIF-1 can induce MAX interactor 1, dimerization protein, which inhibits the transcriptional activity of MYC by competing for MAX and represses MYC target genes [103] such as peroxisome proliferator-activated receptor gamma coactivator 1-beta in human pVHL-null kidney cancer cell lines [104] or ornithine decarboxylase in multiple human cancer cell lines [105]. Second, HIF1A directly inhibits MYC transcriptional activity by DNA-binding site competition. For instance, HIF1A displaces MYC binding from the promoter of cyclin-dependent kinase inhibitor 1A (*CDKN1A*, *p21cip1*) and upregulates the expression of p21 (CDKN1A) in human HCT116 colorectal carcinoma cell line [106]. HIF1A also competes against MYC for binding to SP1, a coactivator of MYC, at the promoters of MYC target genes such as MutS homolog 2 (*MSH2*), *MSH6*, and nibirin, which encode DNA repair proteins, in human HCT116 colorectal carcinoma cell line [107,108] and the E-type prostanoid receptor in human HCA-7 colon cancer cell line [109]. Third, several studies have shown that HIF-1A promotes proteasomal degradation of MYC under chronic hypoxia conditions [104,110–112].

In contrast to HIF1A, HIF2A promotes MYC activity [37]. Overexpression of HIF2A enhances SP1 activity and promotes MYC-driven interleukin 8 expression in human microvascular endothelial cells [113]. HIF2A also enhances MYC activity in human pVHL-null kidney cancer cell lines and primary mouse embryo fibroblasts [102,114]. Consistently, HIF2A deletion has been shown to reduce MYC transcription in human pVHL-null kidney cancer cells implanted in mice [115]. HIF2A promotes MYC activity by stabilizing the MYC/MAX complex [104,116]. Importantly, HIF2A-induced stabilization of MYC/MAX heterodimer is much stronger than HIF1A-mediated degradation of MYC in human cancer cells [104,110–112], leading to MYC activation under hypoxia [116].

MYC upregulates HIF1A proteins although there are no MYC consensus sequences in the promoter of the *HIF1A* gene (Figure 1) [117–130]. The post-transcriptional regulation of HIF1A is responsible for the induction of HIF1A via MYC. For example, transient knockdown of MYC downregulates HIF1A protein levels in multiple human myeloma cells [117]. Overexpression of MYC in human colon cancer and esophageal cancer cells promotes the expression of HIF1A at the post-transcriptional level [118,119]. Overexpression of MYC significantly stabilizes HIF1A and enhances HIF1A accumulation under both normoxic and hypoxic conditions in human normal immortalized mammary epithelial cells and breast cancer cells [120]. Accumulation of HIF1A by MYC leads to the induction of HIF1A targets and is required for MYC-induced anchorage-independent cell growth and proliferation [120]. Mechanistically, MYC prevents HIF1A degradation by reducing HIF1A binding to the pVHL complex, although it increases the level of pVHL complex components [120]. Further, MYC promotes pVHL SUMOylation while repressing its ubiquitination, thereby inhibiting HIF1A ubiquitination and proteasomal degradation [121]. Besides hypoxia, HIF1A expression can be increased via oxygen-independent mechanisms under certain normoxic conditions such as ROS and nitrogen species. MYC increases mitochondrial oxidative phosphorylation and generation of ROS [122]. An increased level of ROS in the mitochondria leads to the stabilization and accumulation of HIF1A by inhibiting prolyl hydroxylase under normoxic conditions [122,123].

Recently, it was shown that MYC induces HIF2A expression as well. MYC has been shown to preferentially bind to the *HIF2A* gene promoter in mouse Sca1C+ cancer stem cells (CSCs) in a MYC-driven mouse T-cell leukemia model and the equivalent ATP-binding cassette superfamily G member 2+ CSC population in human acute lymphoblastic

lymphoma, and activate HIF2A expression [124]. HIF2A regulates stem cell function by inducing the expression of octamer-binding transcription factor 4 [125] and AlkB homolog 5, an m6A demethylase that demethylates *Nanog* mRNA and increases Nanog expression [126]. In fact, the stem cell factors Nanog and SRY-box 2 facilitate MYC-mediated induction of HIF2A, playing a critical role in stem cell renewal and tumor stemness [127].

To date, there is limited literature on the effect of SP1 on *MYC* gene expression or the effect of MYC on *SP1* gene expression. However, Parisi et al. [128] identified a functionally distinct signature for strong dual MYC/SP1 sites in various gene promoters. This finding indicates that although SP1 and MYC do not greatly influence each other's expression transcriptionally or post-transcriptionally, there is a distinct mechanism by which they collaborate to regulate the transcription of specifically selected sets of target genes regulated by both SP1 and MYC.

Overall, these data suggest that there is a positive activation loop of HIF-1 (HIF1A) and SP1, which mostly occurs through induction of the transcriptional activity of the *HIF1A* gene via SP1 and that of the *SP1* gene via HIF-1 (Figure 2). HIF-1 negatively regulates MYC through post-transcriptional mechanisms, and MYC activates HIF-1 through post-transcriptional mechanism. Interestingly, unlike HIF-1 and MYC, there is a positive activation loop of HIF-2 and MYC, which occurs via the combination of both transcriptional and post-transcriptional mechanisms. By contrast, there does not seem to be a direct effect of SP1 on *MYC* transcription or of MYC on *SP1* transcription, although SP1 and MYC collaborate to transcriptionally regulate their target genes.

## 4. Collaboration of SP1, HIF-1, and MYC in Transcriptional Regulation of Their Target Genes

SP1, HIF-1, and MYC interact with each other either transcriptionally or post-transcriptionally and modulate the activity of each other, which demonstrates that there is some collaboration of these TFs in the execution of their activities. However, since SP1, HIF-1, and MYC are first and foremost TFs, their more important collaborations take place when these TFs modulate transcription of their target genes.

Many studies have investigated the mechanisms underlying how SP1 and HIF-1 collaborate in transcriptional regulation of their target genes. One example is the detailed study of the effect of SP1 and HIF-1 on the promoter activity of the human erythropoietin receptor gene [129]. That study showed that the binding of SP1 and HIF-1 to their binding sites in the promoter additively increases the transcriptional activity of the promoter. Another example is the detailed study on regulation of the human retinoic acid receptor-related orphan receptor alpha 4 (*RORalpha*) gene by the interaction between HIF-1 and SP1 [130]. In that case, it was shown that the binding sites for HIF-1 and SP1 in the promoter of this gene are situated closely to each other, and that HIF-1 functionally interacts with SP1 [130]. It was also shown that the HIF2A/SP1/HDAC4 network is involved in transcriptional activation of the human coagulation factor VII gene promoter [131]. Although HIF2A instead of HIF1A is involved in this case, these data suggest that the complex network of HIF1A/HIF2A/SP1/HDAC4 exists, as there is a link between SP1 and HIF1A via HDAC4 (Figure 2D) [95].

The collaboration of HIF-1 and MYC in transcriptional regulation of their target genes has already been described in the previous section. As aforementioned, since HIF-1 and MYC do not modify the expression of each other transcriptionally, the interaction between HIF-1 and MYC occurs either post-transcriptionally (HIF-1 usually suppresses MYC while MYC usually activates HIF-1) or through their collaboration to regulate the expression of their target genes. As an example of HIF-1 modulating the MYC-regulated transcription of genes, for instance, HIF-1 inhibits MYC-dependent induction of the transcriptional activity of the human *CDKN1A* gene promoter via a HIF1A–MYC mechanism [106]. This involves functional antagonism of the transcription repressor MYC via protein–protein interactions. This mechanism is independent of HIF1A DNA binding and transcriptional activity; instead, HIF1A displaces MYC from binding to the *CDKN1A* promoter. A similar

mechanism also works for regulation of the human *MSH2* gene promoter [107]. In this case, neither HIF1A nor MYC binds directly to the *MSH2* promoter. Rather, both HIF1A and MYC discretely interact with the constitutively bound TF SP1 on the *MSH2* promoter, whereas HIF1A dominates SP1 binding in hypoxia by competing with MYC. As a result, SP1 acts as a molecular switch by recruiting HIF1A for the hypoxic repression of *MSH2*. This mechanism is a good example of how HIF-1 can suppress rather than induce gene expression under hypoxia. In addition, this mechanism also shows the diversity of how HIF-1, SP1, and MYC collaborate to control the transcriptional activity.

There is no evidence to suggest that SP1 and MYC directly affect the transcription of each other. However, the collaboration of SP1 and MYC in the regulation of their target genes has been well described in the literature [132–137]. Among the genes whose transcription is regulated by the collaboration of MYC and SP1, there are various genes involved in the regulation of CSCs such as telomerase reverse transcriptase (*TERT*), *BMI1*, cluster of differentiation 133 (*CD133*), and *CD147* [134–137]. These genes are often upregulated in cancer. In fact, most of the genes involved in the regulation of CSCs are regulated by HIF-1 as well [138–141]. Hence, these data indicate that the genes involved in the regulation of CSCs are in most cases regulated by SP1, HIF-1, and MYC. Since CSCs possess 'stemness' properties, which are reflected in their capacity to self-renew and generate differentiated cells that contribute to tumor heterogeneity [142,143], the contribution of CSCs has fundamental importance in the development of cancer; therefore, the eradication of CSCs is crucial for the success of anticancer therapy. As aforementioned, SP1, HIF-1, and MYC are all participants of cancer regulatory networks. The fact that the genes involved in the regulation of CSCs are all controlled by SP1, HIF-1, and MYC indicates that the very reason why these TFs are important participants of cancer regulatory networks might be because they regulate CSCs.

Figure 3 shows the promoters of the human *TERT*, *BMI1*, *CD133*, and *CD147* genes, which contain consensus sequences for SP1, HIF-1, and MYC [136,138–141,144–146]. Either SP1, HIF-1, or MYC bind to their respective consensus sequences in the promoters of these genes and induce transcriptional activity [136,138–141,144–146]. The promoter of the *CD133* gene is very complex, and HIF1A binds to the ETS-binding sites rather than the HRE [136,141]. Interestingly, there are only a small number of HREs, whereas there are often clusters of SP1-binding sites. The HRE and SP1-binding sites are often situated closely together, which suggests that SP1 and HIF-1 collaborate to regulate the transcription of these promoters in a similar manner to that of the *RAR Related Orphan Receptor A* gene promoter [130]. The consensus sequences of the HRE are similar to those of the E-box to which MYC binds [147–149]. Hence, the HRE (or E-box) can provide the point of interaction between HIF-1 and MYC. Regarding the *TERT*, *BMI1*, and *CD147* genes, MYC binds to the HRE (or E-box) in their promoters and controls expression of these genes [137,150,151], whereas in the case of the *CD133* gene, MYC binds somewhere in the vicinity of the CpG islands (which have SP1 consensus sequences) in its promoter and controls expression of the gene [136]. As seen by the great structural differences between the *CD133* gene promoter and those of others, there is some diversity in how SP1, HIF-1, or MYC controls the genes involved in the regulation of stem cells.

Based on the current knowledge about the transcription factors SP1, HIF-1, and MYC, the following conclusions can be drawn. First, as described in Section 2, all HIF-1, SP1, and MYC are deeply involved in cancer-related cellular mechanisms including metabolism, angiogenesis, anticancer immunity, and regulation of TME. Importantly, as described in Section 3, HIF-1 and SP1 usually induce the expression of each other while HIF-1 suppresses the expression of MYC and MYC induces that of HIF-1. This indicates that HIF-1 and SP1 can cooperatively activate cancer-related cellular mechanisms while the relationship between HIF-1 and MYC regarding the regulation of cancer-related cellular mechanisms can be variable depending on the context. Second, the CSC-related genes, which have fundamental importance in oncogenesis, are all positively regulated by HIF-1, SP1, and MYC at the transcriptional level (Figure 3). Overall, these results suggest that inhibitors for

HIF-1 and SP1 likely induce anticancer effects in cooperation by suppressing the activity of cancer-related cellular mechanisms (including the mechanisms underlying CSC regulation) while using MYC inhibitors as anticancer drugs requires some cautions.

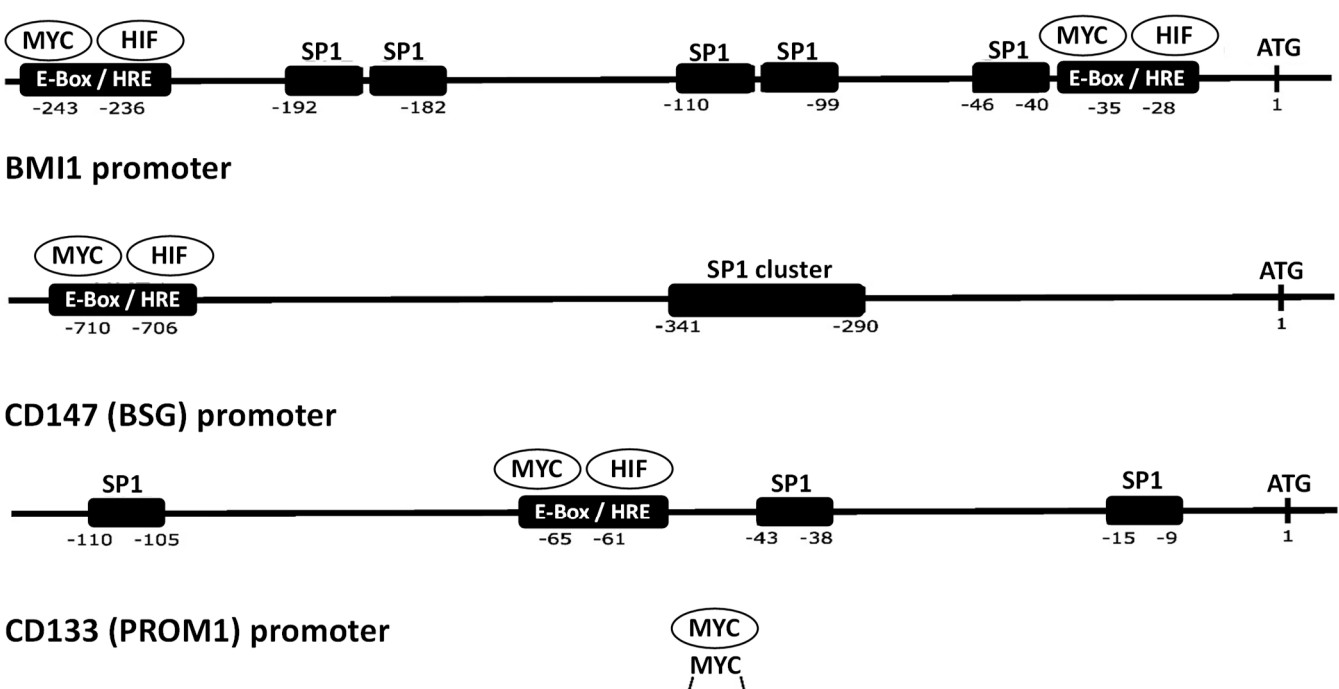

**Figure 3.** Locations of SP1 and HIF-1 consensus sequences in the promoters of genes which are involved in the regulation of CSCs. Schematic of the locations of SP1 and HIF-1 consensus sequences in the promoters of human *TERT*, *BMI1*, *CD147*, and *CD133* genes, which are all involved in the regulation of CSCs. The nucleotide numbers are numbered from the ATG (translation initiation) sites. HIF-1 binds to the ETS consensus sequence in the *CD133* promoter, whereas HIF-1 binds to the HRE in other promoters. The nucleotide sequences of the HRE are similar to those of the E-Box, which is a consensus sequence for MYC binding. The objects with descriptions of 'MYC' or 'HIF' in the circles indicate the TFs. The MYC TF, designated by the 'MYC' in the circle, binds to the E-Box or the MYC consensus sequences in the vicinity of CpG islands (SP1 consensus sequences), whereas the HIF-1 TF, designated by the 'HIF' in the circle, binds to the HRE or the ETS.

**5. Implications of the Interactions and Collaborations of SP1, HIF-1, and MYC in the Development of Anticancer Drugs, and Their Future Perspectives**

*5.1. SP1, HIF-1, and MYC as Targets for Potential Anticancer Therapies*

There are many lines of evidence indicating that SP1, HIF-1, and MYC play important roles in the development of cancer [1–9]. Hence, these TFs must be among the major targets of potential anticancer therapies. Many inhibitors of SP1 and HIF-1 have been developed [11,12]; however, currently an adequately efficient inhibitor of MYC is not available due to the extreme difficulty of designing MYC inhibitors specifically targeted against cancer [15]. Thus, it is unknown how well putative MYC inhibitors work as anticancer agents. In the end, it might turn out that a putative MYC inhibitor is key for efficiently eliminating cancer. On the other hand, a putative MYC inhibitor might have strong side effects due to its broad influence over the normal essential activities of its target

genes. In fact, to some extent, this has been suggested by the results of clinical trials of mithramycin, which is considered to be an SP1 inhibitor but also inhibits transcription of the *MYC* gene [152]. Mithramycin is rarely used as an anticancer drug due to its strong side effects [153]. Although it is not known whether these side effects are derived from the activity of mithramycin in inhibiting *MYC* transcription, this result suggests that MYC inhibition might cause too many side effects.

The design of Omomyc may be used as a guide for the development of effective MYC inhibitors [84]. The action of Omomyc is different from that of many other MYC inhibitors, which have the ability to reduce *MYC* expression by gene knockout or RNA interference. Omomyc selectively targets MYC protein interactions: it binds C-MYC and N-MYC, MAX and MIZ-1, but does not bind MAD or select HLH proteins. Specifically, it prevents MYC binding to promoter E-boxes and the transactivation of target genes while retaining MIZ-1-dependent binding to promoters and transrepression. Clinical trials to date have indicated that the side effects of Omomyc are mild and well tolerated unlike many other MYC inhibitors [154]. In addition, recent findings about the important roles of MYC overexpression in activating super-enhancers, which are often deregulated in cancer, suggest that the points of interactions between super-enhancers and MYC could be more specific targets for designing anticancer drugs than MYC proteins [81]. Recent progress in MYC-related research indicates that MYC inhibitors that are more specifically targeted against cancer-related genes than conventional MYC inhibitors, might be developed in the future.

Currently, there are no anticancer drugs with strong enough activity to eradicate cancer in many patients. One way to compensate for this limitation of anticancer drugs is to combine several drugs to strengthen the anticancer activity of each single drug. Since inhibitors of SP1 and HIF-1 are already available, it is feasible to combine them. As described in Section 2.3, there are many genes that play crucial roles in cancer development and whose expression is regulated by both SP1 and HIF-1. In addition, it was also shown that the genes involved in the regulation of CSCs, which have fundamental importance in cancer development, are regulated by HIF-1, SP1, and MYC (Figure 3). The degree of dependency of the transcription of these genes on either SP1 or HIF-1 can differ from one gene to the other. However, it can be expected that inhibiting the binding of both HIF-1 and SP1 to the promoters of genes whose expression is regulated by both HIF-1 and SP1, should lead to stronger reduction of the expression of these genes than inhibiting binding of either HIF-1 or SP1 alone. Furthermore, since SP1 and HIF1A induce the expression of each other as mentioned above (in other words, positive feedback regulation works for the expression of SP1 and HIF1A; Figure 2), inhibiting the activity of both SP1 and HIF-1 can be expected to downregulate the expression of both TFs far more efficiently than inhibiting the activity of either TF alone. Therefore, it makes sense to combine SP1 inhibitors with HIF-1 inhibitors as potential drugs to be used for combinational anticancer therapy. This concept is shown as a scheme to illustrate what could potentially happen when cancers are treated with a combination of both HIF-1 and SP1 inhibitors (Figure 4). Recently, it was shown that the interaction of SP1 and HIF1A modulated the behavior of cancer cells in a hypoxic microenvironment and promoted cancer development [155]. These data suggest that combination treatment of SP1 and HIF1A inhibitors could effectively disrupt the interaction of SP1 and HIF1A and inhibit cancer development in a hypoxic microenvironment, and support the benefit of using combination treatment of SP1 and HIF1A inhibitors.

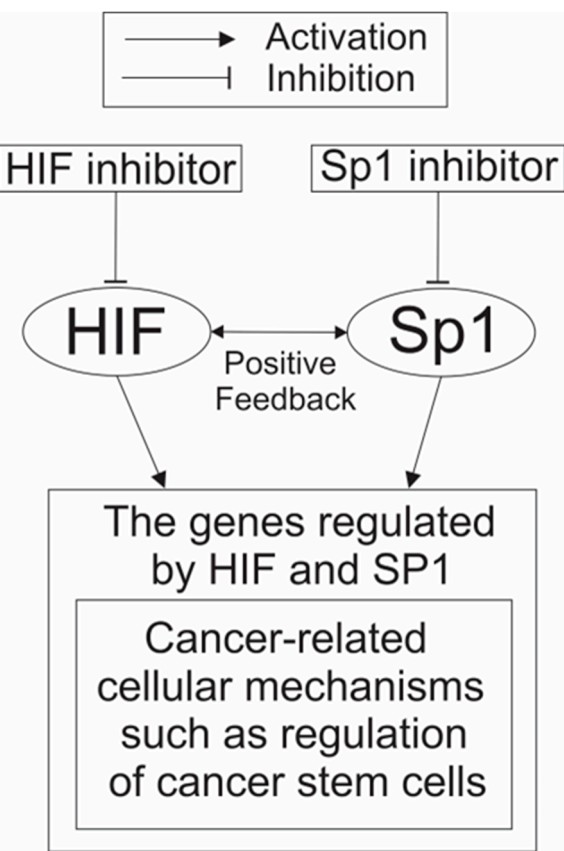

**Figure 4.** Scheme to illustrate what could happen when cancers are treated with a combination of HIF-1 and SP1 inhibitors. There are intricate mechanisms underlying the collaboration of HIF-1 and SP1 to regulate cancer-related cellular mechanisms. HIF-1 and SP1 usually induce the expression of each other (as described in Section 3) and activate their target genes in corporation (as described in Section 4). Therefore, it is expected that combination treatment of HIF and SP1 inhibitors can specifically suppress the activity of cancer-related cellular mechanisms.

### 5.2. Tetramethyl-O-Nordihydroguaiaretic Acid as an Anticancer Drug

Tetramethyl-O-nordihydroguaiaretic acid ($M_4N$) is an anticancer drug candidate that has been studied for many years [156–172]. This compound reversibly inhibits the binding of SP1 to its DNA consensus sequence thereby functioning as an SP1 inhibitor unlike mithramycin which is an irreversible inhibitor of SP1 [156]. $M_4N$ also decreases HIF1A content in cancer cells under hypoxia [159], indicating that it functions as a dual inhibitor of both SP1 and HIF1A. Importantly, $M_4N$ essentially has no strong side effects [171], suggesting that the reversible inhibition of SP1 and HIF1A bindings to their target genes might not cause strong toxicity. $M_4N$ also downregulates the expression of proteins, such as B-cell lymphoma 2/adenovirus E1B 19kDa protein-interacting protein 3, X-linked inhibitor of apoptosis protein, survivin, VEGF, and lactate dehydrogenase A [161,163,164,169,170], and suppresses energy metabolism [162,168]. The function of these proteins as well as energy metabolism are associated with the activities of SP1 and HIF-1 [34,38–44]. $M_4N$ can markedly induce cellular stress in cancer cells [163,169]. The activity of SP1 and HIF-1 in protecting cells from cellular stress has been previously described [72–74], and thus $M_4N$ as an SP1 and HIF-1 dual inhibitor likely suppresses this activity. However, the stress induced by $M_4N$ does not seem to be strong enough to induce significant tumoricidal activity, as $M_4N$ treatment alone has not been shown to induce anticancer activity strong enough to eliminate cancer (although it slows cancer growth) in various xenograft mice and many patients [163,165,166,169,171]. The most compelling data showing the efficacy of $M_4N$ as an

anticancer drug is a clinical trial that was conducted in India in the 1990s treating patients with oral squamous cell carcinoma [172].

However, it might be sufficient to make cancer cells vulnerable against a second anticancer drug that is able to induce strong cytocidal effects, as shown by the synergistic induction of anticancer activity with the combination treatment of $M_4N$ with a second anticancer drug in various xenograft mouse experiments [163,165,166,169]. It was also shown that combination treatment with $M_4N$ suppresses energy metabolism and oncometabolites in addition to inducing strong cellular stress, making it a potentially impressive anticancer treatment [163,169]. In addition, $M_4N$ induces interleukin 21 and enhances B cell-mediated humoral immunity in tumor-bearing mice [173]. It has also been shown that $M_4N$ can downregulate oncometabolites such as lactate and 2-hydroxyglutarate, which suppress activity of the immune system in cancer cells [169], and thus can activate immune reactions via the downregulation of oncometabolites. Overall, these data showed that $M_4N$ could make the TME unfavorable for cancer cells to thrive due to its effect to activate the immune system. The roles of SP1 and HIF-1 in regulating the immune system has already been described in the literature [45–64], which suggests that $M_4N$ induces its effects on the immune system via its activity as a dual inhibitor of SP1 and HIF-1.

### 5.3. A Possible Usage of the Combination Treatment of HIF-1 and SP1 Inhibitors as an Anticancer Therapy

Since $M_4N$, which inhibits both SP1 and HIF-1, does not have strong enough anticancer activity to eliminate cancer in most patients [171], combination treatment of SP1 and HIF-1 inhibitors is also likely not able to induce strong enough anticancer activity to eliminate cancer. However, the results showing that $M_4N$ can induce anticancer activity synergistically in combination with a second anticancer drug [163,165,166,169] suggest that the combination treatment of SP1 and HIF-1 inhibitors could significantly strengthen the anticancer activity of a third anticancer drug. Other than TFs such as SP1, HIF-1, and MYC, there are various additional factors, particularly lncRNAs and miRNAs [174], which play important roles in the development of cancer, as predicted by systems analyses of the regulatory networks in cancer [1–3]. All these factors can naturally be targets of anticancer drugs. To increase the anticancer efficacy of HIF-1 or SP1 inhibitors, combination treatments of HIF-1 and/or SP1 inhibitors with inhibitors of these other factors (e.g., certain lncRNAs and miRNAs as well as MYC), which function as parts of the cancer regulatory networks, may need to be considered as well. In this regard, Omomyc with its selective targeting against MYC-related genes is a particularly attractive candidate to combine with HIF and SP1 dual inhibitors such as $M_4N$ [84].

### 5.4. Roles of HIF-1, SP1, and MYC in Normal Cells and the Potential Toxicity of Anticancer Drugs Targeted on These Transcription Factors

One of many reasons why anticancer drugs do not work well in the clinics is that most anticancer drugs have strong toxicity to normal cells as well as having an effect on cancer cells. As a result, cancer patients treated with these drugs often experience strong side effects and cannot tolerate the drugs until cancers are totally eliminated. Therefore, good anticancer drugs need to specifically target cancer cells without causing strong toxicity to normal cells. Keeping that in mind, the question becomes: how potentially toxic are anticancer drugs that target HIF-1, SP1, or MYC for normal cells?

First, HIF-1 is a master regulator of hypoxic signals so that it is required mainly under hypoxic conditions only [34,35]. Thus, although activation of HIF-mediated signaling mechanisms results in various benefits to cancer cells and the upregulation of HIF-1 is often found in various tumors, only a moderate amount of the expression of HIF-1 is presumably needed to maintain the health of normal cells. For this reason, it would not cause too many difficulties to develop safe and effective inhibitors for HIF-1 as anticancer drugs. Second, SP1 is involved in important normal cellular functions such as cell proliferation and cellular differentiation [5]. However, importantly, although SP1 is essential for normal tissues during embryogenesis and adulthood, it is less important during adulthood than

embryogenesis [5,30,31]. Meanwhile, SP1 is often upregulated in cancers [10,32,33]. This suggests that SP1 inhibitors with only moderate inhibitory activities could possibly incur negative impact on cancer cells without inducing strong adverse effects on normal cells. As mentioned above, mithramycin, a well-known SP1 inhibitor with some activity to inhibit MYC, has strong side effects [152,153]. However, a newly developed mithramycin analogue EC-8042 showed fewer side effects than mithramycin [175], which suggested that there were ways to improve safety of SP1 inhibitors. Moreover, $M_4N$ which only reversibly inhibits SP1 binding to SP1-target genes showed few side effects if any [171]. Overall, this suggests that safe and effective SP1 inhibitors without strong side effects are feasible. Finally, MYC is involved in some of the most important functions in normal cells [75–78]. Therefore, it would be very difficult to eliminate all side effects from any potential MYC inhibitors. In this sense, Omomyc with its selective targeting against MYC-related genes can potentially be a good start to develop safe and effective MYC inhibitors as anticancer drugs [84]. Largely, this analysis suggests that there are potential methods to develop safe and effective combination treatments with HIF inhibitors and SP1 inhibitors.

## 6. Conclusions and Future Directions

HIF-1, SP1, and MYC, which function as master regulators of cancer, interact with each other and modulate the expression of many genes whose functions are associated with the development and maintenance of cancer, such as the genes involved in the regulation of proliferation, CSCs, metabolism, angiogenesis, stress response, and metastasis (as described in Sections 1 and 2). In addition, SP1, HIF-1, or MYC modulates the TME, such as the immune system and the production of oncometabolites, in such a way to facilitate tumor development (as discussed in Section 2). For this reason, inhibitors of HIF-1, SP1, and MYC should be able to work as anticancer agents. Currently, many inhibitors of HIF-1 and SP1 are available, although there are no efficient inhibitors of MYC available due to the extreme difficulty in developing them [15]. In this sense, Omomyc might become a breakthrough for the future development of MYC inhibitors [84]. Although many inhibitors of either HIF-1 or SP1 have some anticancer activity, none of them have activity strong enough to eradicate cancer in the majority of patients. This has led to the idea to combining HIF-1 inhibitors with SP1 inhibitors to improve the anticancer activity of each drug.

It has been shown that $M_4N$ [156–172], a newly developed anticancer drug candidate that inhibits both HIF-1 and SP1, modulates the expression of various genes whose promoters are controlled by SP1 and HIF-1. $M_4N$ also suppresses energy metabolism in cancer and induces humoral immunity in the TME. Clinical trials of $M_4N$ have shown some anticancer activity, but this activity is not strong enough to eradicate cancer in most patients. However, combination treatments of $M_4N$ with various second drugs have been shown to synergistically improve anticancer activity in xenograft mouse studies. The conclusions obtained from the data on $M_4N$ suggest that, while a combination treatment of an HIF-1 inhibitor with an SP1 inhibitor might improve the anticancer efficacy of the drugs to some extent without curing most cancer patients, a combination treatment of HIF-1 and SP1 inhibitors with a third appropriately selected anticancer drug might significantly improve the anticancer activity of the third drug.

In addition, as discussed above, it is realistic to think developing safe and effective inhibitors for HIF-1 and SP1 is achievable. In fact, $M_4N$ with activities as a dual inhibitor for HIF-1 and SP1 does not cause strong side effects [171]. Since we have not tested the safety and efficacy of the combination treatment of $M_4N$ with the second anticancer drug for human patients yet, we do not know how safe and effective this combination treatment can be. Therefore, clinical trials of $M_4N$ (or the combined treatment of HIF-1 and SP1 inhibitors) with various second anticancer drugs are urgently needed. Although it has been shown by numerous mouse xenograft studies that $M_4N$ can synergistically induce anticancer activity with many different second anticancer drugs, the optimal selection of these second drugs has not been established yet. As described, Omomyc is potentially an interesting choice

as a second drug [84]. This line of studies also needs to be carried out together with the clinical trials of the combination treatments suggested above.

**Author Contributions:** Conceptualization, K.K. and R.C.C.H.; methodology, K.K. and R.C.C.H.; validation, K.K. and R.C.C.H.; formal analysis, K.K. and R.C.C.H.; resources, K.K. and T.L.B.J.; data curation, K.K. and R.C.C.H.; writing—original draft preparation K.K. and R.C.C.H.; writing—review and editing, K.K., T.L.B.J and R.C.C.H.; supervision, R.C.C.H.; project administration, K.K., T.L.B.J and R.C.C.H.; funding acquisition, R.C.C.H. All authors have read and agreed to the published version of the manuscript.

**Funding:** This study was supported by grants from the National Institutes of Health (R01DE12165), 806 Biocuremedical, LLC, and the Dorothy Yen Trust (P 690-C25-2407) to RCH. The funders had no role in conceptualization, design of the study, interpretation of the data, decision to publish, or preparation of the manuscript.

**Institutional Review Board Statement:** Not applicable.

**Informed Consent Statement:** Not applicable.

**Data Availability Statement:** Not applicable.

**Conflicts of Interest:** The authors have declared that no conflict of interest exists. However, for full disclosure, we acknowledge that Ru Chih C. Huang is a principal inventor of several Johns Hopkins University patents on M$_4$N.

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
