# Peer review of "Interaction and Collaboration of SP1, HIF-1, and MYC in Regulating the Expression of Cancer-Related Genes to Further Enhance Anticancer Drug Development"

_cimb, doi:10.3390/cimb45110580_

Round 1
Reviewer 1 Report
Comments and Suggestions for Authors
In this manuscript, the authors discuss the strong collaboration between three important transcription factors Myc, SP1, and HIF1. The review aims to understand how the TFs work together in cancer progression and to search for strategies to fight cancer by inhibiting these factors.
The review is well written even if the topic is complex.
The authors should add a Conclusion and Discussion paragraph to comment on all the data mentioned in their review and propose how to cure cancer. The use of myc inhibitors is not simple, as already known, and targeting SP1 produces a lot of side effects, considering that this TF is active both in normal and in cancer cells. The hypoxic environment of cancer probably ensures that only HIF is more active in cancer than in normal cells. This could be a good starting point to discuss the topic of this manuscript. The authors should highlight if there is a clear difference between HIF, SP1, and Myc actions in normal and cancer cells and deeply discuss how the use of their inhibitors (alone or in combination) represents a good choice to fight cancer.
Author Response:
We removed the paragraphs regarding to the analysis of gene regulation by HIF, SP1, and MYC, according to the suggestions from another reviewer. We think that since the roles of HIF, SP1, and MYC in the regulation of the genes which assist cancer development are well investigated, the further analysis of gene expression is not essential for our argument. As discussed in the manuscript, since HIF and SP1 can activate each other, the co-treatment of HIF and SP1 inhibitors should greatly suppress the expression of cancer-related genes including the genes which regulate cancer stem cells.
Instead, according to your suggestion, we added some paragraphs discussing the potential side effects related to inhibitors for HIF, SP1, and MYC based on the roles of these transcription factors in normal cells. We are developing M4N as a potential anticancer drug. This substance is a dual inhibitor for both HIF and SP1 and mostly nontoxic to the patients in the clinical settings. The important feature of M4N as an SP1 inhibitor is that it only reversibly inhibits binding of SP1 to its target DNAs. This might be the reason why M4N does not cause strong side effects. Interestingly M4N, which can suppress expression of various genes and cellular functions which are controlled by HIF and SP1, can synergistically induce anticancer activity with various second anticancer drugs with variable pharmacological activities. We propose a potentially effective anticancer treatment based on these observations and added the paragraph for the conclusion to the manuscript.
Reviewer 2 Report
Comments and Suggestions for Authors
The presented manuscript is devoted to the review of the action of three transcription factors SP1, HIF-1 and MYC. The authors propose the hypothesis that these factors are master regulators of oncogenesis. The first section presents general data on the factors SP1, HIF-1 and MYC, as well as a review of literature data on the cooperation of these factors in the regulation of gene expression. Next, the authors' bioinformatic analysis of ChIP-seq data for the three factors is presented. The article concludes with a discussion of the potential use of inhibitors of the indicated factors as antitumor therapies.
Unfortunately, the presented factor co-occurrence analysis is inadequate. An analysis using a single bioinformatics tool with an unclear algorithm and input data is presented. Conclusions are drawn only on the basis of a very vague classification of terms. The main conclusion cannot be called valid. In order to obtain a reliable conclusion, it is also necessary to analyze transcriptomic data, analyze normal and cancer cells, cells with impaired function of the studied factors, etc.
The ideas of the authors are undoubtedly interesting and important. However, much more in-depth multifaceted analysis is required to reach the conclusion put in the title of the article. The manuscript as submitted cannot be recommended for publication.
Detailed comments:
1) A common observation for all literature cited is that it is unclear whether data are presented for humans or other species?
2) The authors use the Harmonizome program. There is no information in the publication about the raw data taken in the analysis - were they human tissues or cell cultures? Normal cells or cancer cells? Human or animal cells?
The algorithm of the analysis is unclear. What is the criterion for inclusion of a gene in the submitted set - is it sufficient that it was identified in only one experiment or in all published experiments? Based on the description in 4.1, one might assume that the program integrates ChIP-seq data. However, this is not the case - a further explanation is given in 4.2. It turns out that the program also analyzes the presence of factor binding sites. The algorithm for filtering the results is thus not completely clear. Publishing the results in this form is unacceptable; a much fuller description of the input data and the analysis algorithm is needed.
Overall, Harmonizome appears to be an excellent tool for preliminary data analysis. However, using this tool as the only tool to draw any conclusions is inadequate.
3) The paper lacks graphical support. Many textual data can be presented graphically. For example, protein structures (Section 2), gene sampling schemes (Section 4.1).
4) There are questions about data presentation. What is Ontogenetic analysis? How does it relate to the more commonly used GO terms analysis? Which terms in the figures are related to oncogenesis? Are the terms related to oncogenesis highly represented or predominant? What are the statistics on the prevalence of oncogenesis terms? This is the key question from which the authors draw the conclusion given in the title of the publication. In general, the factors studied regulate a wide range of genes. Undoubtedly, among them there are those involved in oncogenesis. How reliably do they predominate?
Unfortunately, no clear answer to these questions can be found in the publication.
What is presented in Fig. 3? What is the meaning of the size and color of the circles and the line between them? The size of the captions to the circles is very small. The list of terms should be sorted based on some criterion.
5) Table 1 is a list of genes with dual regulation. How complete is this list? Is there no bias in its compilation? The criteria for creating this list are even more unclear than the results of using the Harmonizome program.
The format of references in Table 1 is unclear - what is S1, H1?
6) The last section discusses the potential use of simultaneous inhibition of SP1, HIF-1 and MYC as antitumor therapy. The only example, M4N, is an inhibitor of Sp1 binding to DNA, but exhibits a large spectrum of other activities.
As a result, the authors do not provide in the publication real examples of therapies aimed at joint inhibition of the above factors. If such a therapy has not been developed yet, what approaches do the authors propose for the development of drugs of the indicated action?
7) The article lacks a conclusion.
8) The text requires revision.
For example, the phrase "genes related to SP1, HIF-1, and MYC" is unclear. What does related mean? Is it under the control of TF or co-expressed with it? More careful use of terms is needed.
Page 8 has an incomplete sentence: To understand the roles of the collaboration of these TFs in the transcriptional regulation of various target genes, particularly in cancer development.
In the caption to Figure 3: There are 216 genes regulated by both HIF-1 and SP1 (A), 263 genes regulated by both HIF-1 and MYC (B), and 6730 genes regulated by both SP1 and MYC (C). - Nothing about D, E, F.
Author Response:
Thank you for some professional criticism about our statistical analysis of gene regulation. We agreed that the data related to the regulation of gene expression by SP1, HIF-1, and MYC were very complex and that it would be too difficult to analyze them by simple methods. According to your suggestions, we removed all the sections regarding to the analysis of gene regulation from the old manuscript and used only the descriptive analysis about the gene regulation by HIF-1, SP1, and MYC based on published literatures for the new manuscript. We think that since the roles of HIF-1, SP1, and MYC in the regulation of the genes which assist cancer development are well investigated, the analysis of gene expression that we did in the previous manuscript is not essential for our argument. As discussed in the manuscript, HIF-1 and SP1 can activate each other and there are various studies indicating that HIF-1 inhibitors can suppress SP1-mediated gene functions while SP1 inhibitors can suppress HIF-dependent gene functions. These studies suggest that HIF-1 and SP1 are dependent on each other in their functions to some extent and that the co-treatment with the inhibitors for both HIF-1 and SP1 should efficiently suppress the expression of the genes controlled by HIF-1 and/or SP1.
According to your suggestion, we modified the chapter 5 and added the chapter 6 as a conclusion and future perspectives. In the chapter 5, according to a suggestion by another reviewer, we also added some paragraphs discussing the potential side effects related to inhibitors for HIF, SP1, and MYC based on the roles of these transcription factors in normal cells. We are developing M4N as a potential anticancer drug. This substance is a dual inhibitor for both HIF and SP1 and mostly nontoxic to the patients in the clinical settings. The important feature of M4N as an SP1 inhibitor is that it only reversibly inhibits binding of SP1 to its target DNAs. This might be the reason why M4N does not cause strong side effects. Interestingly M4N, which can suppress expression of various genes and cellular functions which are controlled by HIF and SP1, can synergistically induce anticancer activity with various second anticancer drugs with variable pharmacological activities. Any systematic studies about the combination treatment of HIF and SP1 inhibitors have not been done so far although there are various studies indicating that HIF inhibitors can suppress SP1-mediated gene functions while SP1 inhibitors can suppress HIF-dependent gene functions, as described above. There have not been any studies attempting the usage of the combination of HIF and SP1 inhibitors with the third anticancer drugs either. So far, as far as we understand, M4N is the only known drug functioning as a dual inhibitor for SP1 and HIF-1. As described above, M4N can promote anticancer activity of various anticancer drugs with variable pharmacological mechanisms. We formulated a potentially effective anticancer treatment based on these observations obtained from the experiments about M4N and added the paragraph for the conclusion to the manuscript at the end of the chapter 6, according to your suggestion.
We replaced most of the phrases including the expression ‘related to’ with other expressions to clarify the meaning of the description, according to your suggestion.
Round 2
Reviewer 2 Report
Comments and Suggestions for Authors
In the corrected version of the manuscript, the authors removed the part related to their own bioinformatics analysis. Thus, the publication in its present form is a review of the literature on the indicated issue. The authors provide information on the transcription factors discussed, and their interactions. At the end, a discussion of potential therapies targeting the discussed transcription factors is offered.
This publication is of interest to experts in the field of oncogenesis and its therapy.
The following comment still remains. The authors do not indicate in the text for which cell species the data are given. For example, the structure of promoters of several genes is given in Figs. 1, 3. Are these human genes? This should be more accurately indicated throughout the text.
After correction of the comment, the manuscript can be recommended for publication.
Author Response:
Thank you for your comments concerning our previous revisions. As suggested by we have added the information about which cell species the data is for greater clarification.